# Conscious perception of natural images is constrained by category-related visual features

Daniel Lindh[1,2,3], Ilja G. Sligte[3,4], Sara Assecondi [1,2], Kimron L. Shapiro[1,2] & Ian Charest[1,2]

Conscious perception is crucial for adaptive behaviour yet access to consciousness varies for different types of objects. The visual system comprises regions with widely distributed category information and exemplar-level representations that cluster according to category. Does this categorical organisation in the brain provide insight into object-specific access to consciousness? We address this question using the Attentional Blink approach with visual objects as targets. We find large differences across categories in the attentional blink. We then employ activation patterns extracted from a deep convolutional neural network to reveal that these differences depend on mid- to high-level, rather than low-level, visual features. We further show that these visual features can be used to explain variance in performance across trials. Taken together, our results suggest that the specific organisation of the higher-tier visual system underlies important functions relevant for conscious perception of differing natural images.

[1] School of Psychology, Hills Building, University of Birmingham, B152TT Birmingham, UK. [2] Centre for Human Brain Health, University of Birmingham, Birmingham, UK. [3] Department of Psychology, University of Amsterdam, Nieuwe Achtergracht 129-B, 1018 WT Amsterdam, The Netherlands. [4] Amsterdam Brain and Cognition, University of Amsterdam, Amsterdam, The Netherlands. Correspondence and requests for materials should be addressed to D.L. (email: p.j.d.lindh@uva.nl)

Along-standing question in cognitive neuroscience is how visual information is transformed from segregated low-level features to fully conscious and coherent representations. Prevailing object recognition models propose that rapid object identification is accomplished by extracting increasingly complex visual features at various stages/locations of the visual stream[1–3]. Objects are first processed through a hierarchy of ventral visual areas where computations evolve from image feature detection, shape and part segmentation, before more invariant, semantic representations of the objects are established[4–6]. Previous research has shown that animate objects are preferably processed in a broad range of perceptual tasks[7]. This led us to question whether or not animacy also has a preferential access to consciousness, and furthermore, if this could also be true for sub-categories within the animate/inanimate distinction.

Animate versus non-animate object processing has been extensively studied, showing distinct processing pathways throughout the visual stream[8]. Behavioural studies have shown that animate objects are more often consciously perceived in rapid serial visual presentations (RSVP)[9–11], more quickly found in visual search[7], elicit faster responses in discrimination tasks[12,13], and animate words are better retained in working memory[14]. Aggregated, these findings point to a preferential visual processing of animate objects, most likely also reflected in the representational organisation of the visual stream[12,13]. However, the animate categorical division contains several sub-categories also known to cluster together, such as scenes in the parahippocampal place area[15], faces in the fusiform face area[16] and body parts in the extrastriate body area[17] (for review see Martin[18]). It remains unclear how such sub-categories also might differ in visual processing. We address this question by testing differences across several categories (i.e. fruits and vegetables, processed foods, objects, scenes, animal bodies and faces, human bodies and faces), known to cluster together throughout the visual stream, in their propensity to conscious access using the Attentional Blink (AB) paradigm[19].

In the AB paradigm, two targets (T1 and T2) are embedded in a rapidly presented stimulus stream (RSVP). The frequently replicated finding is a reduced ability to report T2 when it is presented in a temporal window of 200–500 ms after a correctly identified T1. This effect disappears when subjects are asked to ignore T1[19], indicating that the fundamental explanation for this effect is attentional rather than perceptual. Most theoretical accounts of the AB suggest a two-stage information-processing model[20,21]. First, both targets are rapidly and automatically processed to a high-level representational stage. This is followed by a capacity-limited second stage, where the percept is transformed into a reportable state (i.e. working memory). Neural findings[22–26] have suggested that the AB arises at the second stage, after semantic processing of the object. This is in contrast to backwards masking, which is known to interrupt feedback loops in early processing[27–29]. Since feedback loops between visual areas are thought to be intact in the AB[26], combined with a behavioural outcome that typically yields a significant number of both correct and incorrect trials, this paradigm is an ideal approach to investigate the bifurcation between conscious and unconscious visual processing.

One potential problem of studying categorical differences is that many categories share low-level scene statistics[30], which also are known to explain behaviour[31]. Consequently, an issue that must be taken into account is how to control for low-level scene statistics in a neurally plausible way. We address this issue by using a Deep Convolutional Neural Network (DCNN)[32] which is designed in a hierarchy encompassing feature representations of increasing complexity, similar to the visual system. Recent studies using DCNNs trained to classify a large corpus of natural images have revealed a significant correspondence between DCNN layers and the visual hierarchical organisation in the brain both using fMRI[33–36], and MEG[5,37]. This makes DCNNs attractive for modelling visual features rather than relying on manually labelling image features without knowing their relevant correspondence to the visual system.

The main question of the current study is if the organisation of the visual system promotes conscious access to certain objects more than to others. A priori, we had two related hypotheses: first we hypothesised that categories will differ in their access to consciousness. Our second hypothesis was that variance in conscious access between image exemplars could be predicted using high-, as opposed to low-, level features derived from the DCNN. These two predictions are consistent with our current understanding of the categorical organisation of the ventral visual stream[4,6,38,39], the high resemblance in representational geometry between the brain and DCNNs[33–37], and theoretical models positing the AB as a disruption of late selection[20,21]. In addition, we explore whether trial-by-trial variance in performance is related to the similarity between the two targets in terms of visual features. We asked whether this relationship has any impact on conscious access and, if so, at what stage of processing do the two targets interact? To test this formally, we used a method called representational sampling, where trials of the AB are constructed with stimuli selected according to their location in DCNN representational geometries. To foreshadow, we show that there are categorical differences in the probability of conscious access. Differences across images are predicted using mid- to high-level visual features. Furthermore, we find a facilitating interaction effect between targets, increasing the probability to recover T2.

## Results

**Experiment 1**. *Differences in AB magnitude as a function of category*: Participants were presented with RSVP, consisting of scrambled masks, and two embedded targets. The targets were selected from a stimulus set of 48 images derived from 8 different categories—fruits and vegetables, processed foods, objects, scenes, animal bodies, animal faces, human bodies and human faces. At the end of each trial, participants were requested to recall the first and the second target (see Fig. 1a). First, we observed a significant AB effect using a two-tailed-dependent $t$-test in T2 performance (T2 performance is always conditional on T1 correct trials; T2|T1) between lags (Lag 2; accuracy M = 0.704, SD = 0.041, Lag 8; M = 0.847, SD = 0.129, $t(18) = -6.427$, $p < 0.001$, see Fig. 2a). We first pooled the images according to animate and inanimate (excluding scenes) objects (see Table 1). Animate and inanimate objects have previously been shown to be differentially affected during the AB[9–11]. Similarly here, a repeated measures $2 \times 2$ ANOVA with lag and animacy as factors showed a main effect of lag (F(1,18) = 34.09, $p < 0.001$, $\eta^2 = 0.654$) and animacy (F(1,18) = 27.72, $p < 0.001$, $\eta^2 = 0.606$) as well as a significant interaction effect (F(1,18) = 45.63, $p < 0.001$, $\eta^2 = 0.606$; see Fig. 2b). Thus, in accordance with previous studies, the AB was less pronounced for animate images. For each sub-category (Table 2), using a repeated measures ANOVA, we observed a main effect of T2-lag (F(1,18) = 42.87, $p < 0.001$, $\eta^2 = 0.704$) and category (F(7,126) = 45.49, $p < 0.001$, $\eta^2 = 0.716$), along with an interaction between category and T2-lag (F(7, 126) = 23.99, $p < 0.001$, $\eta^2 = 0.571$). Beyond the expected AB effect, the interaction effects reveal that different categories exhibit different AB magnitudes (ABM; difference in performance between lag 8 and lag 2). Separate AB effects were tested by contrasting lag 8 and lag 2 performance within each category using a two-tailed-dependent $t$-test (Fig. 2c)—fruits and vegetables ($t(18) = 6.912$, $p < 0.001$),

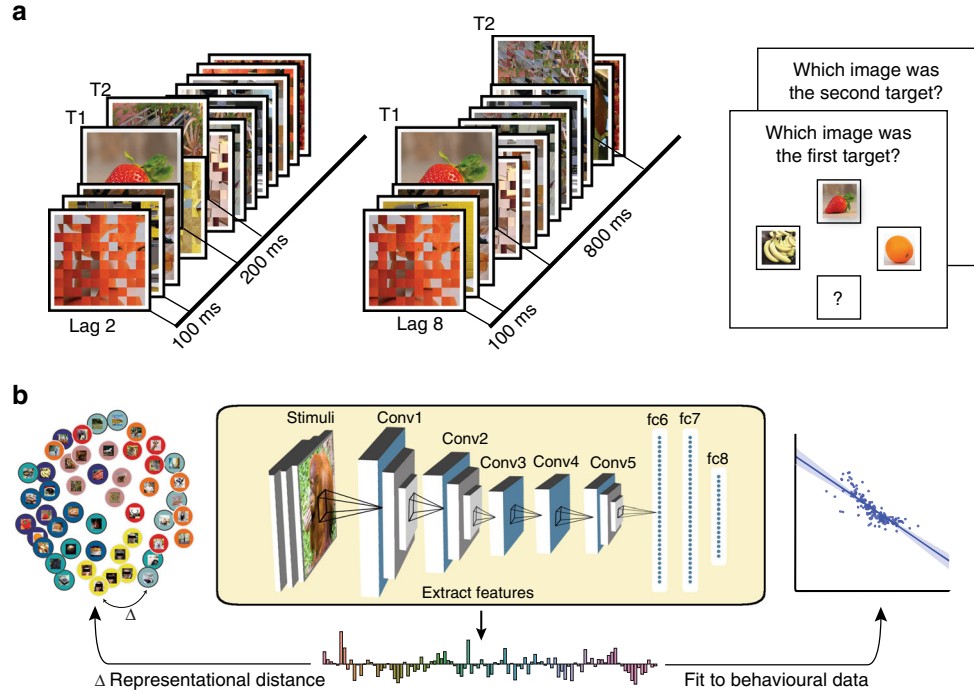

**Fig. 1** Modulating conscious access using the Attentional Blink Paradigm. Due to copyright reasons, all photos except for the faces (which were photographed by one of the authors but have been anonymised) have been replaced by representational images. Eye regions are occluded above in the images to protect privacy but were not occluded in the experiment. **a** We presented a rapid serial visual presentation to participants, with two targets (T1 and T2) following each other within a stream of distractors. On the left, the second target (T2) is shown 200 ms after the first target (T1), and on the right, 800 ms after the T1. In every trial, participants had to detect and later recall both T1 and T2 targets. **b** We used a deep convolutional neural network (DCNN; yellow insert; 5 convolutional layers and 3 fully connected layers) to model the stimulus representational geometries (left) and predict our participants' behaviour (right). The visual stimuli were fed into the DCNN, providing a hierarchical representation for each image. These unit activations were then analysed layer-by-layer and used to predict behaviour

processed foods ($t(18) = 6.748$, $p < 0.001$), objects ($t(18) = 3.003$, $p = 0.004$), scenes ($t(18) = 8.073$, $p < 0.001$), animal bodies ($t(18) = 5.259$, $p < 0.001$), animal faces ($t(18) = 2.712$, $p = 0.007$), human bodies ($t(18) = 1.162$, $p = 0.13$), human faces ($t(18) = 2.632$, $p = 0.008$).

*Mid and high-level image features explain ABM variance*: For each image we extracted unit activations from all the layers throughout an AlexNet DCNN (see Methods). For the convolutional layers, we averaged over the spatial domain to obtain feature activations. It is important to note that this DCNN was trained on classifying objects into categories from a different set of images than those presented in our experiment, and at no point was trained on the AB. To increase the generalization of the model fits to the test data, we selected informational features through a variance thresholding approach. The feature selection was done by calculating the variance across samples in the training data (important to note that the test data was never part of the feature selection) and remove features with near-zero variance from both training and test data. The remaining feature activations were then applied to a cross-validated linear regression model aimed at predicting each image's ABM. From these predicted ABMs, we can compute in each participant the mean absolute error (MAE). For significance testing, we permuted the image labels, repeated the cross-validated linear regression model, and computed the average MAE across subjects. We repeated this permutation procedure 3000 times to estimate the distribution of MAE under the null hypothesis that our image labels are interchangeable. We then compared our observed MAE (averaged across subjects) to this null distribution, and obtained $p$-values. We were able to significantly (Bonferroni-corrected alpha = 0.00625) predict the ABM using features

derived from layer conv4 (MAE M = 0.19, STD = 0.04, $p = 0.003$), conv5 (M = 0.179, STD = 0.04, $p < 0.001$), fc6 (M = 0.159, STD = 0.033, $p < 0.001$), fc7 (M = 0.1593, STD = 0.033, $p < 0.001$), and fc8 (M = 0.191, STD = 0.048, $p < 0.001$). To see whether one layer had significantly lower error than any other layer, we tested the MAE for each pair-wise comparison of layers across subjects with a two-sided-dependent $t$-test. In Fig. 3b, we show a summary of this result, where we find that Layer 7 (Fig. 3c) has a significantly lower error than layer 1 (mean difference = −0.21, $t(17) = −6.14$, $p < 0.001$), layer 2 (mean difference = −0.15, $t(17) = −7.8$, $p < 0.001$), layer 3 (mean difference = −0.16, $t(17) = −10.83$, $p < 0.001$), layer 4 (mean difference = −0.18, $t(17) = −5.8$, $p < 0.001$) and layer 8 (mean difference = −0.18, $t(17) = −5.17$, $p < 0.001$).

*Shared image features between targets predict performance*: In addition to predicting the ABM for each image, we sought to better understand the trial-by-trial differences in the AB. For each trial, we correlated the two targets (T1 and T2) based on their features (Pearson correlation, Fig. 3b) to obtain a T1-T2 similarity measure within each layer. We then averaged the similarity for all hit and miss trials for each participant and tested the difference for each layer using a two-tailed dependent $t$-test. Our test revealed a significantly higher representational similarity between targets in hit-trials compared to miss-trials for layer conv2 (Hit; similarity M = 0.375, SD = 0.008, Miss; M = 0.354, SD = 0.014, $t(18) = 4.967$, $p < 0.001$, Cohen's $d = 1.761$), conv3 (Hit; M = 0.329, SD = 0.010, Miss; M = 0.299, SD = 0.016, $t(18) = 6.273$, $p < 0.001$, Cohen's $d = 2.130$), conv4 (Hit; M = 0.257, SD = 0.009, Miss; M = 0.244, SD = 0.012, $t(18) = 3.505$, $p = 0.003$, Cohen's $d = 1.258$), conv5 (Hit; M = 0.131, SD = 0.007, Miss; M = 0.119, SD = 0.011,

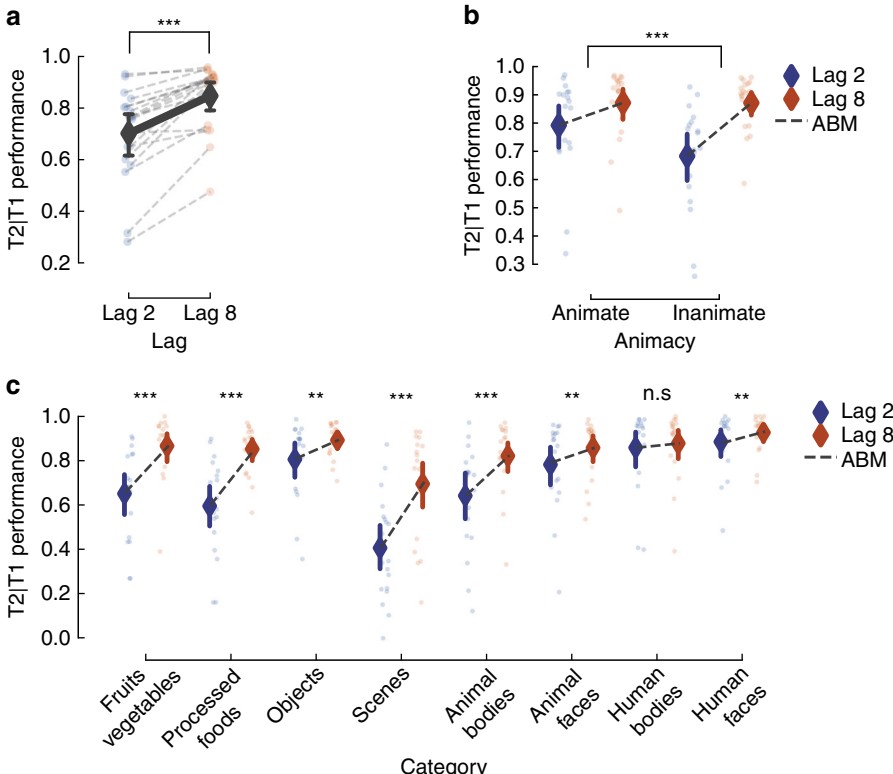

**Fig. 2** Animate objects elicit weaker attentional blink. **a** The accuracy in detecting the second target conditional on having detected the first target for lag 2 and lag 8. Individual dots represent the mean performance for each subject, bold dots represent the mean performance across subjects, and error bars indicate 95% confidence interval around the mean in all plots. **b** Performance plotted separately for animate and inanimate T2 targets. Attentional Blink Magnitude (ABM) is defined as the difference in performance between lag 8 and lag 2. Asterisks indicate significant difference in ABM between animate and inanimate. **c** T2 performance for each category separately. Asterisks indicate an ABM significantly different from zero. Two-tailed-dependent $t$-test *$p < 0.05$, **$p < 0.01$, ***$p < 0.001$

### Table 1 Mean and SDs for T2 performance for each category

| Category | Mean (Lag 2) | SD (Lag 2) | Mean (Lag 8) | SD (Lag 8) | N |
|---|---|---|---|---|---|
| Fruits vegetables | 0.651 | 0.199 | 0.867 | 0.139 | 19 |
| Processed foods | 0.595 | 0.214 | 0.853 | 0.110 | 19 |
| Objects | 0.806 | 0.173 | 0.893 | 0.079 | 19 |
| Scenes | 0.406 | 0.234 | 0.695 | 0.237 | 19 |
| Animal bodies | 0.642 | 0.232 | 0.822 | 0.159 | 19 |
| Animals faces | 0.782 | 0.197 | 0.858 | 0.134 | 19 |
| Human bodies | 0.859 | 0.179 | 0.879 | 0.153 | 19 |
| Human faces | 0.886 | 0.133 | 0.927 | 0.085 | 19 |

### Table 2 Mean and SDs for T2 performance for animacy

| Animacy | Mean (Lag 2) | SD (Lag 2) | Mean (Lag 8) | SD (Lag 8) | N |
|---|---|---|---|---|---|
| Animate | 0.792 | 0.171 | 0.872 | 0.126 | 19 |
| Inanimate | 0.683 | 0.185 | 0.871 | 0.097 | 19 |

$t(18) = 3.311$, $p = 0.004$, Cohen's $d = 1.233$), fc6 (Hit; M = 0.023, SD = 0.002, Miss; M = 0.018, SD = 0.004, $t(18) = 4.009$, $p = 0.001$, Cohen's $d = 1.520$), fc7 (Hit; M = 0.026, SD = 0.003, Miss; M = 0.021, SD = 0.005, $t(18) = 3.189$, $p = 0.005$, Cohen's $d = 1.093$), fc8 (Hit; M = 0.139, SD = 0.013; Miss; M = 0.104, SD = 0.022, $t(18) = 6.134$, $p < 0.001$, Cohen's $d = 1.864$; Fig. 4b). This suggests that the ongoing visual processing of T1 can lower the

conscious access threshold for T2, if T2 shares visual features with T1. This was true for all layers except for layer 1.

**Experiment 2**. *Constructing AB trials using representational sampling*: The finding that T1-T2 similarity influences T2 performance prompted us to design a follow-up study. We sought to investigate the causal effect of target-target similarity by manipulating the targets' category and feature similarity. We developed a procedure called representational sampling, which first characterises a variety of stimulus response profiles, and samples a subset of stimuli tailored for our experiment. We used unit activations from layer 5 (see methods for rationale) of the DCNN as stimulus response profiles. We measured these unit activations on 250 images, derived from ImageNet[40], to yield 16 images as our T2s; in turn chosen to represent four categorical groups equally (mammals, insects, vehicles, and furniture). For each image we then selected two T1s based on category (same or different) and similarity within layer 5 (similar or dissimilar), resulting in eight T1s per T2. This allowed us to examine the specific contribution of high-level feature similarity and category membership separately. We presented these four conditions to 24 new participants in an AB task similar to that of Experiment 1.

Table 3 shows the group means of T2 performance for each of the four conditions. The probability of correctly reporting T2 was the highest when T1 came from the same category and had similar visual feature activation in layer 5 of the DCNN (M = 0.849, SD = 0.097). In contrast, the lowest probability of correctly reporting T2 was observed when T1 came from a different category and was dissimilar (M = 0.741, SD = 0.123). A 2 × 2 (Category by Similarity) repeated measure ANOVA showed a

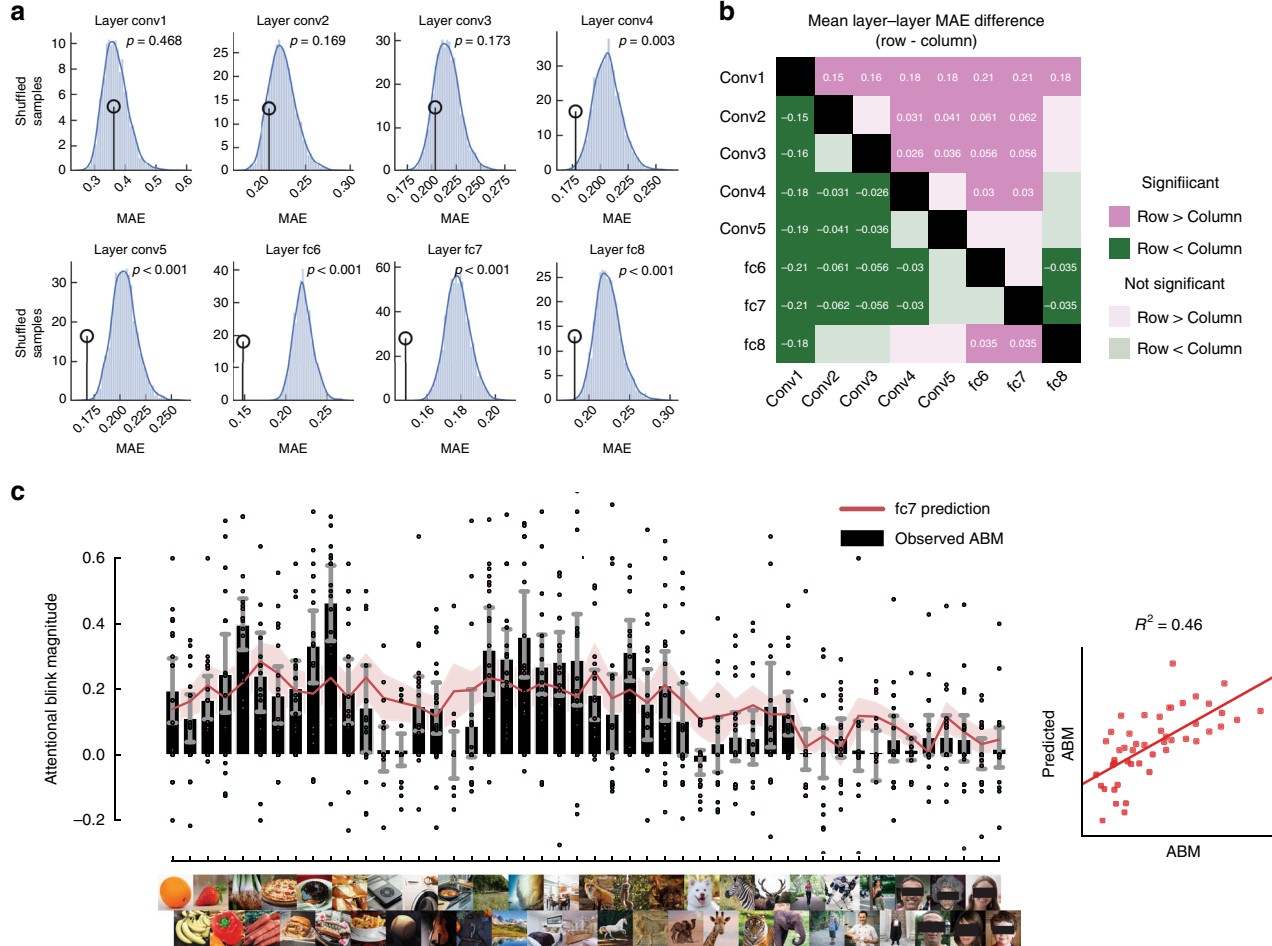

**Fig. 3** DCNN activation units predict attentional blink magnitude. **a** Permutation test distributions. Histograms show the mean absolute error (MAE) after averaging the prediction across participants with randomised image labels. Circles point to the observed MAE. The Bonferroni-corrected alpha value for 8 tests is $p < 0.00625$. **b** Layer by layer comparisons of MAE. Comparisons are done row-wise, where green indicates a lower MAE, or better fit, in comparison to the corresponding column. Only significant (Bonferroni corrected) comparisons are denoted with mean difference in MAE between comparisons. **c** ABM per image. Due to copyright reasons, all photos except for the faces (which were photographed by one of the authors) have been replaced by representational images. Eye regions are occluded in the images to protect privacy but were not occluded in the experiment. Black bars indicate the observed Attentional blink magnitude (ABM), red line is the average predicted ABM based on features from Layer fc7 (which outperformed all other layers, see panel **b**). Individual dots represent individual participants and error bars indicate the 95% confidence interval. Layer fc7 explained 46% of the variance observed. The inset panel shows the average predicted ABM on the y axis, and the average observed ABM per image, on the x axis

significant main effect for both category ($F(1,23) = 20.68$, $p = < 0.001$, $\eta^2 = 0.473$) and similarity ($F(1,23) = 45.468$, $p = < 0.001$, $\eta^2 = 0.664$), as well as an interaction effect ($F(1,23) = 5.413$, $p = 0.029$, $\eta^2 = 0.191$). The larger effect size for the similarity factor indicates that visual features over semantic relevance determine behaviour.

## Discussion

We investigated the effect of category membership and image features on conscious access using natural images in the Attentional Blink[19] paradigm (Fig. 1a, b). By testing images spanning several categories we first show a clear division in performance between animate and inanimate objects, where animate objects reveal a reduced AB caused by the processing of the T1 (Fig. 2b), in line with previous reports[9,10]. We further show that this bias is not only expressed between this super-ordinate division, but also extends to various sub-categories. Using a DCNN to model the stimulus visual features, we show that mid- and high-level features in natural images (Fig. 3) regulate the AB magnitude. In addition, we show that target-target similarity (Figs. 4 and 5) interacts with target selection, providing a mechanistic

explanation of the AB phenomenon and of conscious access in object recognition.

Previous studies have shown differences between categories in the AB, most extensively between animate and inanimate objects[9–11,41]. The animacy bias in visual processing has been attributed to evolutionary relevance, as opposed to visual expertise, reflected in its importance for ancestral hunter-gatherer societies (The animate monitoring hypothesis)[42]. Evidence for this hypothesis comes from a wealth of behavioural studies showing that animate objects are more quickly and more often detected in different types of attentional tasks[7,42]. Likewise, animate and inanimate objects are distinctly represented throughout the ventral visual stream[8,43], which has been argued to be an evolutionary phenomenon and not contingent on visual experience[44]. In our current study, we find that the AB magnitude (ABM – performance difference between Lag-8 and Lag-2) is larger for inanimate objects, similar to Guerrero and Calvillo[10]. The finding by Guerrero and Calvillo has been contested by Hagen and Laeng[11] who showed that animate objects are simply reported more often, but that the ABM is unaffected. Our results argue against the findings of Hagen and Laeng and, more

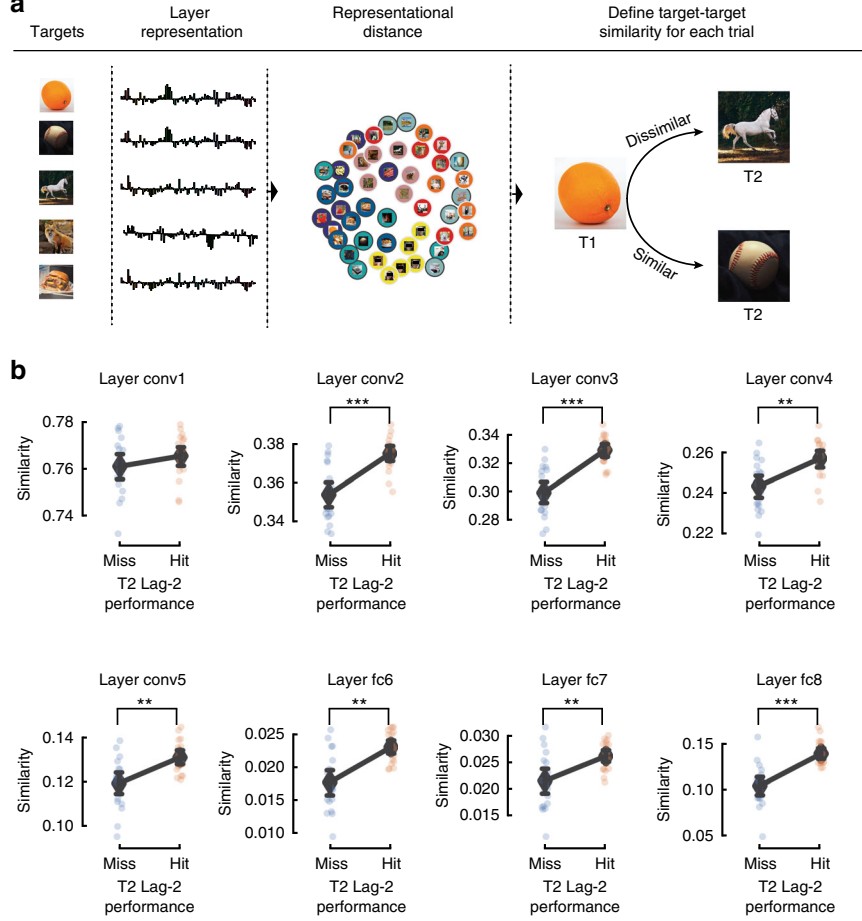

**Fig. 4** DCNN representational distance and target similarity explain trials of the AB. **a** Depiction of analysis procedure. For each layer, DCNN representations are extracted for each image. These feature activations were then compared for all image pairs (Pearson correlation), to estimate the similarity between pairs. Due to copyright reasons, all photos except for the faces (which were photographed by one of the authors) have been replaced by representational images. **b**) Mean similarity between T1 and T2, based on feature activations of each layer, for lag-2 missed and hit trials separately. Separate dots represent single subjects. The mean similarity across subjects is represented by a large black diamond and black bars denote 95% confidence interval. Two-tailed dependent *t*-test *$p < 0.05$, **$p < 0.01$, ***$p < 0.001$

| Table 3 Mean and SDs for T2 performance in experiment 2 | | | | |
|---|---|---|---|---|
| **T2\|T1** | | | | |
| **Category** | **Similarity** | **Mean** | **SD** | **N** |
| Same | Similar | 0.85 | 0.10 | 24 |
| | Dissimilar | 0.81 | 0.09 | 24 |
| Different | Similar | 0.82 | 0.08 | 24 |
| | Dissimilar | 0.74 | 0.12 | 24 |

importantly, reveal that differences in AB magnitude exist in a myriad of sub-categories. Here we examine a significant number of categories, which are known to cluster throughout the visual cortex. We show a high variance in the effect of the AB across categories (Fig. 2c), implying that distinctive sub-categories have special privilege in the path to conscious access. One possible mechanism for categorical differences in conscious access can be related to the findings of Carlson et al.[12], who showed that animate objects that are neurally coded as more animate (as assessed by a decoding scheme) in the human analogous of inferior temporal cortex (hIT) are more quickly categorised as animate in a speeded discrimination task. Translated to our task, this would mean that certain categories are more distinctly represented, with less representational overlap to other images, leading to more

robust processing of these categories. It is important to note that by looking at the differences between Lag-8 and Lag-2, effectively baselining each image with its own Lag-8 performance, our results cannot be explained by differential effects of masking. Importantly, this implies a dissociation between attentional relevance and conscious access, since it would be reasonable to assume that attentional relevance would affect Lag-2 and Lag-8 equally.

The finding that the ABM varies across categories (Fig. 2c) is hard to interpret without properly examining image features of different complexities. Many semantic categories share low-level statistics[30,31] and, without delving further than categorical membership, one cannot disentangle at which level of processing the differences occur. The prediction of ABM across visual objects achieved by modelling DCNN unit activations from the mid to late layers explained a large proportion of AB variance across images (~46% of the variance in layer fc7, Fig. 3c). This implies that the bottleneck produced by the AB is due to late visual processing and probably reflects the particular categorical organisations within higher-tier visual areas. This relationship between neural representation of images and behavioural outcomes is supported by recent work showing that the particular representational organisation in late visual areas predicts certain behavioural measures, such as reaction time[12,13,45]. This 'conceptual' approach to conscious access promotes a more

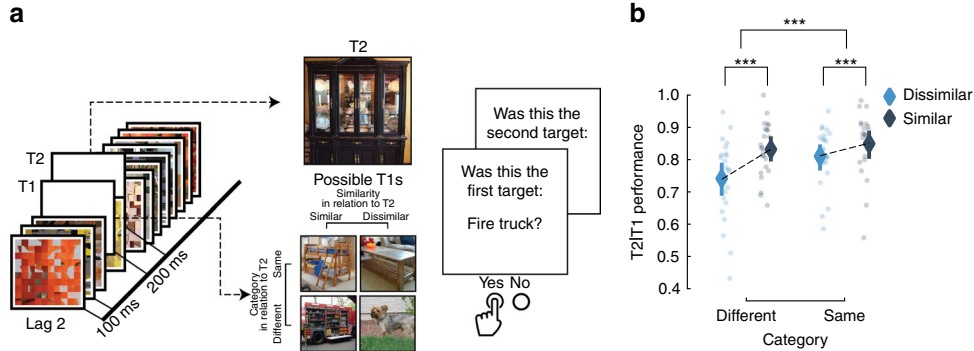

**Fig. 5** Target similarity between T1 and T2 explains T2 performance. **a** Representational sampling was used to construct trials of experiment 2. Each of the sixteen T2s were either preceded by a T1 from the same/different category and similar/dissimilar in representational space within layer 5 of the DCNN. To ensure that participants did not use low-level statistics (such as colour) when reporting the targets, we switched the response menu to a semantic task. **b** Behavioural results from experiment 2. Our results show that features similarity explain a significant portion of T2 performance. Individual dots correspond to individual subjects. Error bars indicate the 95% confidence interval. Statistics were performed using a repeated measures ANOVA (see results). $*p < 0.05$, $**p < 0.01$, $***p < 0.001$

fundamental view to how visual consciousness might operate by focusing on the organisation of the visual system rather than on top-down mechanisms.

Our experiments further enabled us to explore the importance of T1–T2 similarity. Only a handful of studies have investigated target-target similarity in the context of AB[9,41,46–48]. In one of the earliest attempts to study target–target similarity and its effect on T2 performance, Awh et al.[46] concluded that similarity between targets is detrimental to T2 reportability. This led to the multiple-resource channel hypothesis (MRCH)[46]. According to the MRCH, two targets (T1 and T2) can be processed in parallel, but only if their visual features are different enough to be processed through distinct feature channels. While a few following studies have corroborated this notion[41,47,48], our study reveals that similarity is beneficial for performance. The difference in results might be explained by the way we define similarity by image features. Previous studies used categorical membership as a proxy to similarity, and thus it is possible that our findings reflect a facilitation effect not found in the previous studies (but see ref. [9]). Importantly, while visual features function as stepping stones toward semantic meaning, it is unclear that such visual features would be maintained in working memory in our paradigm. Task-relevant similarity (i.e. the semantic content stored in working memory necessary to successfully carry out the task) between targets has been shown to be key for inducing a larger blink[48]. We would argue that the visual features within the DCNN models processes that precede working memory representations. As such, the target–target similarity rather enhance visual processing of T2, leading to a more probable recovery. The combined findings of all these studies highlight a relatively unexplored aspect of AB, where the relationship between the targets might play a significant role in explaining many AB phenomena. Further questions could be explored using a combination of brain measures to determine representational similarity within subjects, which might potentially also explain individual differences in performance.

In conclusion, we present compelling evidence that there are categorical differences in conscious access in object recognition. Specifically, we present findings that attribute differences in conscious access between image exemplars to difficulties in representational readouts of features in higher-tier visual areas. This visual feature-related bias is reflected in a stable functional organisation, where fined-grained category distinctions have a larger impact on conscious access than previously believed. Moreover, we point to a more dynamic way in which the context (i.e. the similarity between T1 and T2) biases the probability for a

target to be consciously perceived. In summary, our findings suggest that object categories and high-level visual features constrain conscious perception of natural images.

## Methods

**Experiment 1.** *Participants*: Twenty participants (19 females; age range: 19–22; mean = 20.1 ± 1.2) were recruited for the study. We excluded two participants due to incomplete data. One additional participant was excluded for the image-by-image analyses due to lack of trials where T2 was correct for one image after filtering for T1 correct. All participants provided and signed informed consent and were rewarded for their time via course credits or financial compensation (at the standard rate of £7/h). All participants had normal or corrected-to-normal vision, and no known history of neurological disorders. The Ethical Review Committee of the University of Birmingham approved the experiment.

*Procedure*: Participants viewed visual objects in a RSVP, and were asked to detect two targets (T1 and T2) embedded into a stream of distractors (Fig. 1a). Following the stream, a response menu was presented for T1, which included the T1 and two foils, and the participant had to identify the target with a button press. A similar response menu was then presented to identify the T2. The foils in the menu always belonged to the same category as the targets (Fig. 1a, right panel).

*Design and stimuli*: Participants were seated 60 cm away from a Stone monitor (60 Hz refresh rate), and stimuli covered 5 degrees of visual angle centrally on a grey background. Stimulus presentation was achieved using the Psychtoolbox extension (version 3[49]) in MatLab 2016b (MathWorks Inc., Natick, USA). Stimuli consisted of 48 images, derived from eight different categories: fruits and vegetables, processed foods, objects, scenes, animal bodies, animal faces, human bodies, and human faces (Fig. 1b). It is important to note that images were displayed in greyscale to reduce performance for human observers. To generate the items used as distractors in the stream, each image was divided into 5 × 5 (25 total) squares. Each square was then inverted and randomly assigned to a new square position. Following a standard AB paradigm[19], each trial started with 300 ms of fixation, followed by a RSVP consisting of 19 images. Each image was presented for 16.7 ms with a stimulus-onset asynchrony (SOA) of 100 ms (Fig. 1a). Embedded into the stream of distractors, two non-scrambled targets (T1 and T2) were presented at two different lag conditions (Lag-2: 200 ms and Lag-8: 800 ms). The T1 was always item 5 in the stream, while T2 was either presented as item 7 (Lag 2) or item 13 (Lag 8). Each participant completed 12 runs (excluding one practice run of 5 trials). Across all runs each image was presented 12 times as T2 for both lags, for a total of 24 repetitions per image, and a total of 1152 trials. All 48 images were presented on an equal number of trials either as T1 or as T2, randomized within blocks with no trial having the T1 and T2 coming from the same superordinate category. Importantly, the same pair of T1 and T2 was always presented in both the Lag-2 and Lag-8 conditions, within the exact same stream of distractor masks in the RSVP trial. Participants had to press one out of three buttons to identify the correct target from the foils, or a fourth button when they missed the target. The two foils came from the same category as the target.

*Deep convolutional neural network (DCNN)*: We employed a DCNN (AlexNet, see Fig. 1c)[32], implemented through Python and Caffe[50], as a model of the visual cortex for extracting hierarchical visual features from our stimuli (we do not intend the use of model here to mean an exact biological model, but merely to approximate the hierarchical architecture that is known to exist in both). We chose AlexNet due to its relative simplicity, compared to more recent DCNNs, and its well-studied relation to the human visual system[33–36,43]. AlexNet consists of eight

layers of artificial neurons stacked into a hierarchical architecture, where preceding layers feed-forward information to the next layer (Fig. 1b). The first five layers are convolutional layers, whereas the last three are fully connected layers. While the fully connected layers (fc6, fc7, and fc8) consist of one dimensional arrays (sizes of 4096, 4096, and 1000 units respectively), the convolutional layers have the dimensionalities of: layer 1 (conv1)—$96 \times 55 \times 55$ (96 features, over $55 \times 55$ retinotopic units), layer 2 (conv2)—$256 \times 27 \times 27$, layer 3 (conv3)—$384 \times 13 \times 13$, layer 4 (conv4)—$384 \times 13 \times 13$, and layer 5 (conv5)—$256 \times 13 \times 13$. For all analyses we averaged the values in the convolutional layers for each image over the spatial dimension, leaving them with the vector length of 96, 256, 384, 384, and 256 respectively. This network was pre-trained on 1.3 million hand-labelled, natural images (ImageNet; Russakovsky et al.[40]) for classification into 1000 different categories (available at http://caffe.berkeleyvision.org/model_zoo.html), reaching near-human performance on image classification[32]. Our test set of 48 images were analysed through the network, and we used the last processing stage of each layer as model output for further analyses. To keep the images as close to the training data as possible, and to avoid distortions of all levels of feature representations, the colour versions of the images were used.

*Analyses of behaviour and image features*: For each image, we calculated mean T2 accuracy at both Lag-2 and Lag-8 across subjects. We then computed ABMs by subtracting Lag 2 mean accuracy from Lag 8 mean accuracy. ABM then becomes a measure of how much the AB time window affects the recall of each image separately. In the interest of quantifying image features, within our DCNN, we extracted unit (neuron) activation patterns for each image from all the layers. For the first five convolutional layers, we averaged the activation over the spatial dimension. These activation patterns were incorporated into a multivariate linear regression model, with the activation patterns from each layer as features in the model to predict each image's ABM within subjects. The prediction pipeline followed a leave-one-image-out procedure (i.e. train on forty-seven images and test on one left out image)— where, based on the training data, the features were thresholded to have a larger variance than 0.15, to remove near-zero-varying features, and later standardised to unit variance with a mean of zero. Our choice of a threshold of 0.15 was arbitrary and had little to no effect when compared to only removing zero variance features. It's important to note that the test data was never part of any feature selection, as this would constitute double dipping. All pre-processing and fitting procedures were implemented using Sci-kit learn[51], for python code see https://github.com/Charestlab/abdcnn.

*Target–target similarity*: We further tested the effect of target-target similarity on conscious access. Here, we go beyond using predetermined categories as a proxy for feature similarity and examined the representational distance between images within a given layer of the DCNN. For each layer we calculated the Pearson correlation between all possible T1–T2 pairs (Fig. 4a). We then averaged the similarity for hit and miss trials separately. This allowed us to test the difference between hit and misses in terms of the relationship between the targets.

**Experiment 2**. *Participants*: We recruited 24 participants (Age—M = 19.38, SD = 0.95, females = 19, males = 5) with normal, or corrected-to-normal, vision. All participants provided and signed informed consent and were rewarded for their time via course credits or financial compensation (at the standard rate of £7/h). The experiment was approved by the ethics committee at the University of Birmingham.

*Procedure and stimuli*: Unless stated otherwise, all procedure and visual presentations were identical to Experiment 1 (Fig. 5a). Sixteen images, a subset of 250 labelled and processed images from the ImageNet database[40], were selected as T2s. The T2s derived from four different categories (mammals, insects, vehicles, and furniture), and each category was uniformly represented in the T2 selections. Similarity between images was determined by their Pearson correlation coefficient within layer 5 of the DCNN. The layer 5 was chosen because it was a high-performing layer in the first study and to still maintain the retinotopic information for an additional analyses not used in this study. To model the layer-wise unit activations for this new set of images, we used the same pre-trained network (AlexNet)[32] as in Experiment 1. For each T2, we selected two similar and two dissimilar images from the same category and any of the other categories as T1. This resulted in eight potential T1s for each T2 in a 2-by-2 factorial design (Similarity × Category) (Fig. 5a). Each condition had the following mean Pearson correlation between T1 and T2, Same category/Similar layer 5 representation (Pearson r M = 0.43, SD = 0.114), Same category/Dissimilar (M = 0.136, SD = 0.113), Different category/Similar (M = 0.337, SD = 0.114) and Different category/Dissimilar (M = −0.056, SD = 0.099). T1 was always placed at position 11, and T2 at position 13 (in a RSVP of 19 items for each trial). Each block consisted of a presentation of each T2 paired with every possible T1, for a total of 128 trials per block divided into 4 runs (32 trials per run). Each participant completed 2 blocks for a total number of 256 trials per session (64 trials per condition).

**Reporting summary**. Further information on research design is available in the Nature Research Reporting Summary linked to this article.

## Data availability
Data associated with this article can be found, in the online version, at https://github.com/Charestlab/abdcnn/. A reporting summary for this article is available as a Supplementary Information file.

## Code availability
Code associated to the manuscript is available at https://github.com/Charestlab/abdcnn/.

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

## Acknowledgements

This work was supported by a European Research Council (ERC) Starting Grant ERC-2017-StG 759432 (to I.C.). We would like to thank Sara Binks and Alfie Brown for their help in collecting the behavioural data from experiments 1 and 2 (respectively), Jasper van den Bosch, and Howard Bowman for comments on the manuscript.

## Author contributions

S.A., D.L., and I.C. contributed to the design of the experiments. D.L. analysed the data. D.L., I.S., S.A., K.S., and I.C. contributed to the writing of the manuscript.

## Additional information

**Competing interests:** The authors declare no competing interests.

