## [Peer Review File · Nature Communications]

Reviewers' Comments:

Reviewer #1:

Remarks to the Author:

The authors used deep convolutional neural networks (dCNNs) as an explanatory mechanism of the attentional blink phenomenon. In the first experiment, the authors demonstrated the the magnitude of the AB (T2-T1 performance) was not the same over different object categories, and that ~46% of the variance of the observed difference could be explained by layer FC7 of an "AlexNet" dCNN. A secondary analysis revealed that the more similar the T2 and T1 targets were (with respect to dCNN layer, in all layers except Conv1), the higher the ABM. Experiment 2 employed a causal test via representational sampling. Each T1 was chosen to be in one cell of the following 2x2: same versus different category, or similar versus different according to Conv5 CNN features. They found that both category membership and CNN-based similarity drove ABM, and that these factors also interacted.

Major comments:

-1- Unclear what the motivation is here

From the first paragraph, the authors do not make the motivation of the study clear. (How conscious access to objects differs as a function of category and visual features). This is stated alternatively on page 3 "if the organization of the visual system promotes conscious access to certain objects more than others". Again, why might this be the case? Without this, it seems like this paper is a solution in search of a problem.

-2- Very strong assumptions about CNNs

An implicit assumption of this work is that the representations in the CNN layers have a strong one-to-one mapping with the workings of the human visual system. Examples of this assumption include noting that the dCNN is a "model of the visual system" (first paragraph, Discussion), or when inferring levels of processing in the visual system from corresponding later layers of the CNN (e.g. "This implies that the bottleneck produced by the AB is due to late visual processing...."). While there is growing evidence of *correlations* between CNNs and human observers, the magnitude of these correlations is not sufficiently strong to be able to accept a CNN as a model biological visual system. It's important to remember that while the engineers creating these systems were inspired by biology, that there are a number of critical differences as well.

Smaller points:

-1- I am confused about the Bonferroni corrections to the results presented at the bottom of page 5. What exactly is being corrected? It doesn't seem to be the eight layers because there is a different corrected alpha for each layer. It doesn't seem to be the number of features in each layer because it seems that the authors are (for example) correcting for 50 features in FC7 when this layer has 4096 features. Please explain.

-2- The color scheme for Figure 3b is likely to be difficult for dichromats to decipher, and will likely not render well if printed in grayscale.

-3- If the monitor refresh was set to 60 Hz, then the image presentation could not be 16 ms, as a single monitor refresh is 16.7 ms.

Reviewer #2:

Remarks to the Author:

Lindh et al. examined whether conscious perception can be explained by hierarchical visual features derived from deep convolutional neural networks (DCNN). The authors utilize the attentional blink paradigm to modulate conscious access of perceptions and find that high-level image features explain the magnitude of attentional blinks. They also show that image similarity in the high-level feature space can explain the attentional blink magnitude for newly sampled images.

The experiments show clear effects, and the representational sampling experiments demonstrate the generalizability of their main findings appropriately. My concern about the current manuscript is that the descriptions regarding experiments and analyses lack some important details. The authors should provide more information regarding the descriptions and analyses to clarify the manuscript.

Specific comments

The authors should specify more details regarding Experiment 1: How many trials/runs/blocks/images/repeats per image did they present?

In the caption of Figure 1, the authors state that the stimuli were shown in grayscale. Is this true and, if so, why? Does this also mean that all of the stimuli—including targets, distractors, and foils—were shown in grayscale? Did they also feed grayscale images to the DCNN to extract features? I am not sure that such a choice is optimal in terms of perception and feature extraction from DCNN. They should provide some more clarification of, and/or justification for this particular choice of stimuli.

The authors should provide more details regarding analysis: How many image-samples did they use for training the model? If I interpret the current manuscript correctly, they used 48 images in a “leave-one-out” manner (I believe that this means “leave-one-image-out”), which indicates that they had only 47 training samples while the number of features was up to 4096. If they performed a linear ordinary least squares regression in this way (without any regularization, presumably), I wonder whether such an ill-posed problem would cause overfitting. How many features did they remove at the preprocessing stage? How did they determine the threshold for removal?

Are the terms “T2 accuracy”, “T2 performance”, and “T2|T1 (performance)” interchangeable? If this is the case, to avoid potential confusion, the authors should use the same term. If not, they should clarify the differences.

Tables 1 and 2: N should be 19, not 19.000.

Figure 3A: Is the y-axis really “Proportion”, rather than the number of shuffled samples?

Figure 3C inset: R^2 should be $R^{²}$ (to the power of 2).

Figure 4B, T2 accuracy: Were they the averages over the trials and the lags? Or did they pick the accuracy

for, say, Lag 2?

Methods/Experiments 1: some titles look confusing; Experimental procedure followed by Procedure. I wonder whether the first one should be Participants.

Reviewer #3:

Remarks to the Author:

The paper by Lindh et al presents a very elegant and clear demonstration that object categories that have distinct representations in the brain, also have differential effect for conscious visual perception.

Overall, I found the paper to be well-written and interesting, and the approaches novel and compelling. I also think the paper will be of interest to the community. The authors clearly demonstrate relationships between AB effects and different object categories, and how DCNN activations can explain aspects of the performance, however, my main concern is that I was left unconvinced we have learnt too much about why different categories have differential effects of conscious access.

Major comments:

1. The central idea here is that high-level visual features are important to conscious perception, and that conscious access is further different across categories. I think this study still leaves me wondering what is it about the categories that creates this effect? The authors nicely show that T1-T2 visual feature overlap impacts conscious access, and so these kind of features are important, but is it the overlap of the features that is the relevant dimension (or category homogeneity), or that animate objects have more features (so layer 5, for example, has more strongly activated nodes for faces vs. vehicles), or some combination. It is true that the authors show that different categories have different AB effects, and that higher image similarity aids perception of T2, but I don't feel the authors explained why the AB is greater for some categories than others. If category-specific activity in the brain is used to explain this, then there are clusters for both animate and inanimate objects in the ventral stream, face and place specific regions, so I am left wondering why some categories are better perceived than others? The categories may have a difference in conscious access but why? Without saying something about this, the main contribution of the paper seems to be that DCNNs can explain AB effects, meaning we have learnt something about DCNNs rather than about conscious access to different objects.

2. On page 8, the authors state that 'features from layer 7 is best suited for explaining the difference of AB effect across images' – and we are referred to figure 3B. However, I would find it helpful if the authors could explain why this is the case more clearly. In the text, layer 5 has the smallest MAE suggesting it is the best, although I think the evidence the authors are drawing on are the row values in figure 3B to justify the claim about layer 7. Reading through a few times, I understand the plots and why the claim is made, but think the authors could add some justification in the text for this point, and add text about what the numbers in Figure 3B are. For example, adding the mean layer-layer difference for each row might help, in addition to saying why some values are missing (presumably a statistical test was conducted?, and if not, claims about a 'best' layer would need some statistical evaluation). This might make it clear that, although layer 5 has lowest MAE, layer 7 has a reduced MAE for the images compared to most other layers.

3. Correlations between T1-T2 similarity and T2 accuracy are conducted on binned data. This seemed strange to me, and unnecessary. Aren't you losing variance in the data by doing this, and shouldn't it be the case that the continuous T1-T2 similarity measures correlate with the continuous T2 accuracies? This would be the obvious thing to do, and so I was left wondering why the authors chose a binning approach? Further, how did they decide the bin sizes, and whether the bins were defined based on T2 accuracies or T1-T2 similarities?

Other comments:

1. In the introduction it is a little unclear what is meant by category. This term is used variously across studies and could be referring to an animal, a dog or even a poodle. As such, it might be worth defining the categories, or what you mean by the word, or using a more specific label (e.g. superordinate category, although I'm not sure that quite fits).

2. It was a little tricky comparing performance for lags 2 and 8 in table 1. Perhaps putting lag 2 and 8 in different columns would make this easier for the reader.

3. I'm not sure Figure 3C is referred to in the paper.

4. In experiment 2, the authors choose stimuli based on the layer 5 activations, and define images as similar or different. This is a very nice idea, but the paper was missing some descriptive statistics for the 4 conditions they defined. What was the T1-T2 similarity for the 4 conditions? Presumably they couldn't create 4 matched conditions, but this data would be good to include.

5. On page 14, the authors state that feature selection was based on removing low variance features. Could they quantify what this means?

Reviewer's comments

Reviewer #1 (Remarks to the Author):

The authors used deep convolutional neural networks (dCNNs) as an explanatory mechanism of the attentional blink phenomenon. In the first experiment, the authors demonstrated the magnitude of the AB (T2-T1 performance) was not the same over different object categories, and that ~46% of the variance of the observed difference could be explained by layer FC7 of an "AlexNet" dCNN. A secondary analysis revealed that the more similar the T2 and T1 targets were (with respect to dCNN layer, in all layers except Conv1), the higher the ABM. Experiment 2 employed a causal test via representational sampling. Each T1 was chosen to be in one cell of the following 2x2: same versus different category, or similar versus different according to Conv5 CNN features. They found that both category membership and CNN-based similarity drove ABM, and that these factors also interacted.

We would like to thank the reviewer for these comments.

Major comments:

-1- Unclear what the motivation is here

From the first paragraph, the authors do not make the motivation of the study clear. (How conscious access to objects differs as a function of category and visual features). This is stated alternatively on page 3 "if the organization of the visual system promotes conscious access to certain objects more than others". Again, why might this be the case? Without this, it seems like this paper is a solution in search of a problem.

Response: We thank reviewer 1 for pointing out that the main motivation behind the report is not made clearly in the first paragraph. Previous research has shown that animate objects are more quickly detected in search tasks, better remembered in working memory tasks and seem to be processed more quickly by the perceptual system. This led us to question whether or not animacy also has a preferential access to consciousness, and furthermore, if this could also be true for sub-categories within the animate/inanimate distinction.

Action: We have added clarifications regarding the motivation of the study in the first paragraph of the revised manuscript (p. 2):

"Previous research has shown that animate objects are preferably processed in a broad range of perceptual tasks (Jackson & Calvillo, 2013). This led us to question whether or not animacy also has a preferential access to consciousness, and furthermore, if this could also be true for sub-categories within the animate/inanimate distinction."

Action: We have also addressed reviewer 1's query as to *why this might be the case* in the Discussion on Pages 11-12.

-2- Very strong assumptions about CNNs

An implicit assumption of this work is that the representations in the CNN layers have a strong one-to-one mapping with the workings of the human visual system. Examples of this assumption include noting that the dCNN is a “model of the visual system” (first paragraph, Discussion), or when inferring levels of processing in the visual system from corresponding later layers of the CNN (e.g. “This implies that the bottleneck produced by the AB is due to late visual processing...”). While there is growing evidence of *correlations* between CNNs and human observers, the magnitude of these correlations is not sufficiently strong to be able to accept a CNN as a model biological visual system. It’s important to remember that while the engineers creating these systems were inspired by biology, that there are a number of critical differences as well.

Response: It is true that the exact mapping between CNNs and the visual stream is still under debate, however, studies have shown that the representational geometries show a remarkable similarity between the two systems, with CNNs outperforming traditional computer vision models in variance explained for human IT data (Khaligh-Razavi & Kriegeskorte, 2014). Importantly, our intention is not to provide further evidence that CNNs are a true likeness of brain function and architecture, but to draw upon the fact that there are hierarchical similarities between the two systems, with more abstract information being processed the further one goes in the system. This provides a means to distinguish low-level from mid-to-high level visual features in a more elegant way than earlier possible. Bearing this in mind, we agree with the reviewer that our use of “model of the visual system” could be interpreted literally as meaning an exact biological model of the visual system (which no current visual model is), so we clarified this in the manuscript.

Action: Clarified in the revised manuscript on page 11:

“Using a DCNN to model the stimulus visual features, we show that mid- and high-level features in natural images (Fig 3) regulate the AB magnitude.”

Action: In the revised manuscript on page 14 we removed the statement that we used the DCNN as a model of the visual system.

Smaller points:

-1- I am confused about the Bonferroni corrections to the results presented at the bottom of page 5. What exactly is being corrected? It doesn’t seem to be the eight layers because there is a different corrected alpha for each layer. It doesn’t seem to be the number of features in each layer because it seems that the authors are (for example) correcting for 50 features in FC7 when this layer has 4096 features. Please explain.

Response: We thank the reviewer for pointing this out. The Bonferroni correction is to accommodate testing the prediction accuracy of each of the 8 layers of the DCNN. We realise that the more common way is to adjust the alpha rather than adjusting the actual p-values (and keep the alpha at 0.05).

Action: We have changed the method of how the Bonferroni corrections are implemented. Instead of correcting the p-values, we are now adjusting the alpha value and reporting the

uncorrected p-values (which is equivalent but hopefully more clear) in Figure 3A. The Bonferroni corrected alpha for eight tests (one for each layer) is $p = 0.00625$.

-2- The colour scheme for Figure 3b is likely to be difficult for dichromats to decipher, and will likely not render well if printed in grayscale.

Response: We thank the reviewer for pointing this out.

Action: We have updated the colour scheme for Figure 3b.

-3- If the monitor refresh was set to 60 Hz, then the image presentation could not be 16 ms, as a single monitor refresh is 16.7 ms.

Response: This was an oversight on our part. Thanks to the reviewer for pointing this out.

Action: We have clarified this in the revised manuscript on page 13.

Reviewer #2 (Remarks to the Author):

Lindh et al. examined whether conscious perception can be explained by hierarchical visual features derived from deep convolutional neural networks (DCNN). The authors utilize the attentional blink paradigm to modulate conscious access of perceptions and find that high-level image features explain the magnitude of attentional blinks. They also show that image similarity in the high-level feature space can explain the attentional blink magnitude for newly sampled images.

The experiments show clear effects, and the representational sampling experiments demonstrate the generalizability of their main findings appropriately. My concern about the current manuscript is that the descriptions regarding experiments and analyses lack some important details. The authors should provide more information regarding the descriptions and analyses to clarify the manuscript.

We would like to thank the reviewer for the useful comments and suggestions.

Specific comments

The authors should specify more details regarding Experiment 1: How many trials/runs/blocks/images/repeats per image did they present?

Response: Each participant completed 12 runs (excluding one practice run of 5 trials). Across all runs each image was presented as T2 12 times for both lags (for a total of 24 repetitions). This has now been clarified in the manuscript.

Action: We provided additional descriptions regarding experiment 1 in the manuscript.

On page 4:

“Participants were presented with a Rapid Stream of Visual Presentations (RSVP), consisting of scrambled masks, and two embedded targets. The targets were selected from a stimulus set of 48 images derived from 8 different categories – fruits and vegetables, processed foods, objects, scenes, animal bodies, animal faces, human bodies and human faces. At the end of each trial, participants had to recall the first and the second target (see Figure 1A).”

On page 13:

“Each participant completed 12 runs (excluding one practice run of 5 trials). Across all runs each image each was presented 12 times as T2 for both lags, for a total of 24 repetitions per image, and a total of 1152 trials”

In the caption of Figure 1, the authors state that the stimuli were shown in grayscale. Is this true and, if so, why? Does this also mean that all of the stimuli—including targets, distractors, and foils—were shown in grayscale? Did they also feed grayscale images to the DCNN to extract features? I am not sure that such a choice is optimal in terms of perception and feature extraction from DCNN. They should provide some more clarification of, and/or justification for this particular choice of stimuli.

Response: As correctly noted by the reviewer, we presented the stimuli in grayscale to our participants. This was done in order to increase the overall difficulty of the task. Indeed, feeding grayscale images to the DCNN to extract features would be a suboptimal choice. For this exact reason, the DCNN was fed the coloured versions of our images. Object colour is a crucial diagnostic feature for DCNNs, and removing the colour information comes with the risk of degrading the higher order representations in the DCNN late layers. We therefore agree with the reviewer that feeding grayscale images to our DCNN would not be the optimal choice. However, only the hierarchical information of the DCNN is relevant for our analysis and conclusions and since our results are based on a cross-validated approach, there is no risk of overfitting of any non-essential features.

Action: We have also clarified that we used grayscale images in the revised manuscript on page 13:

“Targets and distractors were presented in grayscale in order to increase difficulty.”

The authors should provide more details regarding analysis: How many image-samples did they use for training the model? If I interpret the current manuscript correctly, they used 48 images in a “leave-one-out” manner (I believe that this means “leave-one-image-out”), which indicates that they had only 47 training samples while the number of features was up to 4096. If they performed a linear ordinary least squares regression in this way (without any regularization, presumably), I wonder whether such an ill-posed problem would cause overfitting. How many features did they remove at the preprocessing stage? How did they determine the threshold for removal?

Response: We agree with the reviewer that there is a great risk of overfitting when using more features than observations. We believe however, that this concern is reduced when projecting the ordinary least square (OLS) training beta weights on the left-out test image to estimate our predicted behavioural scores. The features of the left-out test image were never seen by the OLS model, and therefore the model could not learn anything about their noise structure. If the model is evaluated on the same data as the training then accuracy could be inflated; however, if the model is evaluated on separate test data as we did, the obtained accuracies are unbiased. In addition, we test for significant predictions using a permutation procedure. Under the null hypothesis that the image labels are interchangeable, overfitting would also bias the null distribution and prevent from finding a significant fit. The fact that our cross-validated prediction error rates are lower than those in the null distribution makes us confident that we are not overfitting. Finally, we employed a feature selection procedure that diminishes the contribution of zero-variance non-informative features in the fit (we threshold the variance to be at least 0.15 for any given feature), reducing the number of features in the training set, and further reducing potential overfitting. This has now been clarified in the manuscript.

Action: we provided further clarifications about our feature selection procedure on page 6. In addition, we clarified our leave-one-image-out cross-validation procedure on page 13-14.

Page 6:

“To increase the generalization of the model fits to the test data, we selected informational features through a variance thresholding approach. The feature selection was done by calculating the variance across samples in the training data (important to note that the test data was never part of the feature selection), and remove features with near-zero variance from both training and test data.”

Page 13:

“The prediction pipeline followed a “leave-one-image-out” procedure (i.e. training on forty seven images and test on one left out image) – where, based on the training data, the features were thresholded to have a larger variance than 0.15 to remove near-zero-varying features, and later standardized to unit variance.”

Are the terms “T2 accuracy”, “T2 performance”, and “T2|T1 (performance)” interchangeable? If this is the case, to avoid potential confusion, the authors should use the same term. If not, they should clarify the differences.

Response: Yes, they are and we understand the confusion. This has now been addressed throughout the manuscript.

Action: We streamlined our terminology throughout the manuscript and now we refer to it all as “T2 performance”, and we now clearly define it in the revised manuscript on page 4:

“First, we observed a significant AB effect in T2 performance (T2 performance is always conditional on T1 correct trials; T2|T1) between lags”

Tables 1 and 2: N should be 19, not 19.000.

Response: thanks for pointing this out.

Action: we have now fixed this in the revised manuscript p. 5.

Figure 3A: Is the y-axis really “Proportion”, rather than the number of shuffled samples?

Response: We thank the reviewer for pointing this out.

Action: We have changed the label for the y-axis on figure 3A.

Figure 3C inset: R2 should be R^2 (to the power of 2).

Response: Again, thanks for this.

Action: we have now made this correction in the revised manuscript on figure 3C.

Figure 4B, T2 accuracy: Were they the averages over the trials and the lags? Or did they pick the accuracy for, say, Lag 2?

Response: We thank the reviewer for pointing this out. Our primary focus for this analysis was to look at items directly following the T1, i.e. lag-2. We have now clarified in the revised manuscript that the results in figure 4B are based on lag-2 T2 performance.

Action: We have clarified that the results in figure 4B are based on lag-2 T2 performance. Also, due to a question by reviewer 3 we also have changed the way the data is analysed in order to reduce confusion. For more information, we refer to our response to reviewer 3.

Methods/Experiments 1: some titles look confusing; Experimental procedure followed by Procedure. I wonder whether the first one should be Participants.

Response: We thank the reviewer for noticing this.

Action: We have changed the name of the heading describing our participants to ‘Participants’

Reviewer #3 (Remarks to the Author):

The paper by Lindh et al presents a very elegant and clear demonstration that object categories that have distinct representations in the brain, also have differential effect for conscious visual perception.

Overall, I found the paper to be well-written and interesting, and the approaches novel and compelling. I also think the paper will be of interest to the community. The authors clearly demonstrate relationships between AB effects and different object categories, and how DCNN activations can explain aspects of

the performance, however, my main concern is that I was left unconvinced we have learnt too much about why different categories have differential effects of conscious access.

We would like to thank the reviewer for their comments on the manuscript and for identifying this crucial point.

Major comments:

1. The central idea here is that high-level visual features are important to conscious perception, and that conscious access is further different across categories. I think this study still leaves me wondering what is it about the categories that creates this effect? The authors nicely show that T1-T2 visual feature overlap impacts conscious access, and so these kind of features are important, but is it the overlap of the features that is the relevant dimension (or category homogeneity), or that animate objects have more features (so layer 5, for example, has more strongly activated nodes for faces vs. vehicles), or some combination. It is true that the authors show that different categories have different AB effects, and that higher image similarity aids perception of T2, but I don't feel the authors explained why the AB is greater for some categories than others. If category-specific activity in the brain is used to explain this, then there are clusters for both animate and inanimate objects in the ventral stream, face and place specific regions, so I am left wondering why some categories are better perceived than others? The categories may have a difference in conscious access but why? Without saying something about this, the main contribution of the paper seems to be that DCNNs can explain AB effects, meaning we have learnt something about DCNNs rather than about conscious access to different objects.

Response:

Our study was designed to investigate categorical differences in conscious access, which leads to our main conclusion. We find that there are clear differences between the categories in behaviour, and that higher-level model representations better predict performance. This is in contrast to low-level features, which are also shared among categories, but do not seem to predict behaviour as well. This leads to the conclusion that the categorical differences we observe are mainly due to differences in representations during late visual processing. The reviewer pose an interesting idea that maybe animate objects have a *richer representation* (i.e. more features) than inanimate objects and thus are more resilient to the AB. Prior to this, we did not have any reason to believe that there would be more features in the animate category. To verify this we measured the mean feature activation for each image in the fc7 layer, see image inserted below.

The figure shows the mean unit activation (in arbitrary units) for all images, with no clear apparent advantage for animate or inanimate objects (an independent t-test contrasting animate (image 25-48) vs. inanimate (image 1-24) objects was also non-significant $p = 0.48$). We appreciate the reviewers idea of looking into *representational richness* and it might be something we want to investigate in future studies.

In order for us to investigate *how* the categorical organisation of visual areas drives the differences in conscious access, we suspect that a carefully designed fMRI project would need to be conducted. Our CNN does not give us a biological precise model of the visual system, but it does enable us to look at hierarchical features in a more realistic way than previously possible. We can only speculate here, based on our data, and expect that follow-up studies will be designed to answer this important question. One possible account stems from the findings of Carlson et al. (2014), who show that animate objects that are neurally coded as “more” animate in the human analogue of inferior temporal cortex (hIT) are processed more rapidly. With reference to our study, this suggests that certain categories are more distinctly represented, i.e., have less representational overlap with other images, leading to more robust processing. The question as to how this particular bias in representational distinctiveness (as shown by Carlson et al.) has emerged throughout the evolution of humankind could be seen as relating to the *animate monitoring hypothesis* (New, Cosmides, & Tooby, 2007). This hypothesis posits that, due to the fact that our species has evolved to monitor animate objects, e.g., to eat and to avoid being eaten, the human visual system has developed mechanisms to allocate attention automatically and rapidly to these objects. Importantly, our results suggest there might be sub-hierarchies within the animate/inanimate categories for which we have developed different adaptations, either through evolution or through life-time exposure.

Action: We have added a passage in the Discussion on page 11 to expound these ideas:

“One possible mechanism for categorical differences in conscious access can be related to the findings of Carlson et al. (2014), who showed that animate objects that are neurally coded as “more” animate (as assessed by a decoding scheme) in the human analogous of inferior temporal cortex (hIT) are more quickly categorised as animate in a speeded discrimination task. Translated to our task, this would mean that certain categories are more distinctly represented, with less representational overlap with other images, leading to more robust processing of these categories.”

2. On page 8, the authors state that ‘features from layer 7 is best suited for explaining the difference of AB effect across images’ – and we are referred to figure 3B. However, I would find it helpful if the authors could explain why this is the case more clearly. In the text, layer 5 has the smallest MAE suggesting it is the best, although I think the evidence the authors are drawing on are the row values in figure 3B to justify the claim about layer 7. Reading through a few times, I understand the plots and why the claim is made, but think the authors could add some justification in the text for this point, and add text about what the numbers in Figure 3B are. For example, adding the mean layer-layer difference for each row might help, in addition to saying why some values are missing (presumably a statistical test was conducted?, and if not, claims about a ‘best’ layer would need some statistical evaluation). This might make it clear that, although layer 5 has lowest MAE, layer 7 has a reduced MAE for the images compared to most other layers.

Response: We thank the reviewer for pointing this out; it is a clarification that we overlooked. There was a discrepancy between how the MAE was calculated for the overall group statistics (layer by layer) and the layer comparisons tests. The layer by layer statistics were previously relying on averaging the predicted ABM values across-subjects, and the MAE was the difference between the group-average prediction and the group-averaged ABM. We have modified this approach, and now compute the MAE in each subject, and average the MAEs across, which is symmetric to the test performed in the layer comparison analysis. The difference between the two procedures is extremely negligible, but we now obtain the same averaged MAE with both approaches, and with layer 7 showing lowest MAE.

Action: We have clarified what our row and column values mean in Figure 3B. We have clarified the description of the permutation test and we have added the statistical results on pages 6-7:

“The remaining feature activations were then applied to a cross-validated linear regression model aimed at predicting each image’s ABM. From these predicted ABMs, we can compute in each participant the mean absolute error (MAE). For significance testing, we permuted the image labels, repeated the cross-validated linear regression model, and computed the average MAE across subjects. We repeated this permutation procedure 3000 times to estimate the distribution of MAE under the null hypothesis that our image labels are interchangeable. We then compared our observed MAE (averaged across subjects) to this null distribution, and obtained p-values. We were able to significantly predict the ABM using features derived from layer conv4 (MAE $M = 0.17$,

STD = 0.045, $p = 0.003$), conv5 ($M = 0.158$, STD = 0.032, $p < 0.001$), fc7 ($M = 0.1489$, STD = 0.03, $p < 0.001$), and fc8 ($M = 0.168$, STD = 0.036, $p < 0.001$, Bonferroni corrected $\alpha = 0.00625$). To see if one layer had significantly lower error than any other layer, we tested the MAE for each pair-wise comparison of layers across subjects. In Fig 3B we show a summary of this result, where we find that Layer 7 (Fig 3C) has a significantly lower error than layer 1 (mean difference = -0.17, $p < 0.001$, $t = 6.669$), layer 2 (mean difference = -0.043, $p < 0.001$, $t = 6.3$), layer 3 (mean difference = -0.043, $p < 0.001$, $t = 5.88$), layer 6 (mean difference = -0.025, $p < 0.001$, $t = 7.12$) and layer 8 (mean difference = -0.02, $p = 0.001$, $t = 3.88$).”

3. Correlations between T1-T2 similarity and T2 accuracy are conducted on binned data. This seemed strange to me, and unnecessary. Aren't you losing variance in the data by doing this, and shouldn't it be the case that the continuous T1-T2 similarity measures correlate with the continuous T2 accuracies? This would be the obvious thing to do, and so I was left wondering why the authors chose a binning approach? Further, how did they decide the bin sizes, and whether the bins were defined based on T2 accuracies or T1-T2 similarities?

Response: We thank the Reviewer for raising an issue that may be raised by others reading our paper. To address this issue we changed to a more appropriate method for showing that T1/T2 similarity is beneficial for conscious report. We now average the similarity between targets within hit and miss trials, and test the difference using a 2-tailed t-test. Additionally, we now report effect sizes in the text.

Action: We have changed Fig 4B and its caption, as well as provided the appropriate changes in the text.

Page 8:

“We then averaged the similarity for all hit and miss trials for each participant and tested the difference for each layer using a two-tailed dependent t-test. Our test revealed a significantly higher representational similarity between targets in hit-trials compared to miss-trials for layer conv2 (Hit; similarity $M = 0.375$, $SD = 0.008$, Miss; $M = 0.354$, $SD = 0.014$, $t(18) = 4.967$, $p < 0.001$, Cohen's $d = 1.761$), conv3 (Hit; $M = 0.329$, $SD = 0.010$, Miss; $M = 0.299$, $SD = 0.016$, $t(18) = 6.273$, $p < 0.001$, Cohen's $d = 2.130$), conv4 (Hit; $M = 0.257$, $SD = 0.009$, Miss; $M = 0.244$, $SD = 0.012$, $t(18) = 3.505$, $p = 0.003$, Cohen's $d = 1.258$), conv5 (Hit; $M = 0.131$, $SD = 0.007$, Miss; $M = 0.119$, $SD = 0.011$, $t(18) = 3.311$, $p = 0.004$, Cohen's $d = 1.233$), fc6 (Hit; $M = 0.023$, $SD = 0.002$, Miss; $M = 0.018$, $SD = 0.004$, $t(18) = 4.009$, $p = 0.001$, Cohen's $d = 1.520$), fc7 (Hit; $M = 0.026$, $SD = 0.003$, Miss; $M = 0.021$, $SD = 0.005$, $t(18) = 3.189$, $p = 0.005$, Cohen's $d = 1.093$), fc8 (Hit; $M = 0.139$, $SD = 0.013$, Miss; $M = 0.104$, $SD = 0.022$, $t(18) = 6.134$, $p < 0.001$, Cohen's $d = 1.864$; Fig 4B)”

Page 14:

“For each layer we calculated the Pearson correlation between all possible T1-T2 pair, we then averaged the similarity for hit and miss trials separately. This allowed us to test the difference between hit and misses in terms of the relationship between the targets.”

Other comments:

1. In the introduction it is a little unclear what is meant by category. This term is used variously across studies and could be referring to an animal, a dog or even a poodle. As such, it might be worth defining the categories, or what you mean by the word, or using a more specific label (e.g. superordinate category, although I'm not sure that quite fits).

Response: We thank the reviewer for pointing out this ambiguity. We agree with the reviewer that the word category can be ambiguous. In our manuscript we define categories as items that group together in the human analogue of inferior temporal cortex.

Action: We now clearly define the categories in the introduction on page 1:

“We address this question by testing differences across several categories (fruits and vegetables, processed foods, objects, scenes, animal bodies and faces, human bodies and faces), known to cluster together throughout the visual stream, in their propensity to conscious access using the Attentional Blink paradigm (AB; Raymond, Shapiro, & Arnell, 1992).”

2. It was a little tricky comparing performance for lags 2 and 8 in table 1. Perhaps putting lag 2 and 8 in different columns would make this easier for the reader.

Response: Thanks for pointing this out.

Action: We have made the proposed changes to table 1 and 2 on page 5.

3. I'm not sure Figure 3C is referred to in the paper.

Response: We thank the reviewer for pointing this out.

Action: We now refer to Figure 3 panel C on pages 7 and 11.

4. In experiment 2, the authors choose stimuli based on the layer 5 activations, and define images as similar or different. This is a very nice idea, but the paper was missing some descriptive statistics for the 4 conditions they defined. What was the T1-T2 similarity for the 4 conditions? Presumably they couldn't create 4 matched conditions, but this data would be good to include.

Response: This is a good point. Thanks. Indeed, there will be on average larger similarities within category than between categories. We have now added the mean similarity and their standard deviations for the four conditions of the second experiment.

Action: We have added further information in the method section on page 15:

“Each condition had the following mean Pearson correlation between T1 and T2 in the layer 5 representation, “Same category/Similar” (Pearson r M = 0.43, SD = 0.114), “Same category/Dissimilar” (M = 0.136, SD = 0.113), “Different category/Similar” (M = 0.337, SD = 0.114) and “Different category/Dissimilar” (M = -0.056, SD = 0.099).”

5. On page 14, the authors state that feature selection was based on removing low variance features. Could they quantify what this means?

Response: We used a variance threshold method of removing near zero-variance (less than 0.15) features. This point was also raised by reviewer 2.

Action: We provided further clarifications about our feature selection procedure on page 6.

Page 6:

“To increase the generalization of the model fits to the test data, we selected informational features through a variance thresholding approach. The feature selection was done by calculating the variance across samples in the training data (important to note that the test data was never part of the feature selection), and remove features with near-zero variance from both training and test data.”

Page 13-14:

“The prediction pipeline followed a “leave-one-image-out” procedure (i.e. training on forty seven images and test on one left out image) – where, based on the training data, the features were thresholded to have a larger variance than 0.15 to remove near-zero-varying features, and later standardized to unit variance.”

Reviewers' Comments:

Reviewer #1:

Remarks to the Author:

I thank the authors for their clear reply and responsiveness to the suggestions of the reviewing team. I am mostly happy with this current revision, but still have a lingering concern over the conceptual status of the dCNN.

Although the workings of dCNNs are still largely "black boxes", there seems to be broad consensus that features in early layers are more reflective of low-level visual features, while later layers reflect more conceptual content. Thus, it's both interesting and puzzling that except for the first dCNN layer, similarity in all dCNN layers is predictive of AB performance (Figure 4). It may therefore be the case that both low-level visual feature similarity as well as conceptual similarity are at play here. I don't think that another analysis is worthwhile, but I would be interested in the authors' view of what type(s) of similarities are at play, and feel that this would deepen the discussion.

Reviewer #2:

Remarks to the Author:

I thank the authors for the clarifications. I was Reviewer #2 and have several remaining concerns.

>> Action: We have also clarified that we used grayscale images in the revised manuscript on page 13:
> "Targets and distractors were presented in grayscale in order to increase difficulty."

I could not find this edit. Also, if the authors presented stimuli to the participants in grayscale and fed the stimuli to DCNN in color, they should describe that. The current manuscript has an ambiguity.

>> the features were thresholded to have a larger variance than 0.15 to remove near-zero-varying features, and later standardized to unit variance.

What was the unit or relative contribution of this 0.15 value? Was this after some pre-normalizations (e.g., the highest variance channel had the variance of 1.0)? What percentages of the feature channels did they eliminate (e.g., 5%, 20%, or 90%)? I do not want to be a method nerd, but the current description is not specific enough to reproduce or make sense of the analysis.

>> Figure 3A: Is the y-axis really "Proportion", rather than the number of shuffled samples?

> Response: We thank the reviewer for pointing this out.

> Action: We have changed the label for the y-axis on figure 3A.

Figure 3A still has "Proportion."

Reviewer #3:

Remarks to the Author:

I feel the authors have responded to my comments, and adjusted the manuscript to addresses them.

Reviewer's comments

Reviewer #1 (Remarks to the Author):

I thank the authors for their clear reply and responsiveness to the suggestions of the reviewing team. I am mostly happy with this current revision, but still have a lingering concern over the conceptual status of the dCNN.

Although the workings of dCNNs are still largely “black boxes”, there seems to be broad consensus that features in early layers are more reflective of low-level visual features, while later layers reflect more conceptual content. Thus, it's both interesting and puzzling that except for the first dCNN layer, similarity in all dCNN layers is predictive of AB performance (Figure 4). It may therefore be the case that both low-level visual feature similarity as well as conceptual similarity are at play here. I don't think that another analysis is worthwhile, but I would be interested in the authors' view of what type(s) of similarities are at play, and feel that this would deepen the discussion.

Response: We thank reviewer 1 for this interesting line of reasoning. Based on the literature and our current knowledge this seem to be dependent on task-relevancy, i.e. the semantic content stored in working memory in order to successfully carry out the task. While its known from earlier studies that task-relevant similarity between T1 and T2 seem to induce larger blinks (e.g. Sy & Giesbrecht, 2009), conversely we found that similarity in visual features (which are not stored directly in working memory as far as we know) have a facilitating effect. While these visual features work as stepping stones towards semantic they are not the end product stored in WM. This finding would not have been possible to find without a relevant model with visual hierarchy, such as a DCNN.

Action: We extended the discussion on this on page 11-12:

“Importantly, while visual features function as stepping stones toward semantic meaning, it is unclear that such visual features would be maintained in working memory in our paradigm. Task-relevant similarity (i.e. the semantic content stored in working memory necessary to successfully carry out the task) between targets has been shown to be key for inducing a larger blink⁴⁷. We would argue that the visual features within the DCNN models processes that precede working memory representations. As such, the target-target similarity rather enhance visual processing of T2, leading to a more probable recovery.”

Reviewer #2 (Remarks to the Author):

I thank the authors for the clarifications. I was Reviewer #2 and have several remaining concerns.

>> Action: We have also clarified that we used grayscale images in the revised manuscript on page 13:
> “Targets and distractors were presented in grayscale in order to increase difficulty.”

I could not find this edit. Also, if the authors presented stimuli to the participants in grayscale and fed the stimuli to DCNN in color, they should describe that. The current manuscript has an ambiguity.

Response: We thank the reviewer for pointing this out again.

Action: We clarified this on page 12 for human observers:

“It’s important to note that images were displayed in greyscale to reduce performance for human observers.”

And page 13 for the DCNN:

“To keep the images as close to the training data as possible, and to avoid distortions of all levels of feature representations, the colour versions of the images were used.”

>> the features were thresholded to have a larger variance than 0.15 to remove near-zero-varying features, and later standardized to unit variance.

What was the unit or relative contribution of this 0.15 value? Was this after some pre-normalizations (e.g., the highest variance channel had the variance of 1.0)? What percentages of the feature channels did they eliminate (e.g., 5%, 20%, or 90%)? I do not want to be a method nerd, but the current description is not specific enough to reproduce or make sense of the analysis.

Response: We thank the reviewer for allowing us to clarify the role of the feature selection. The main function of the variance thresholding is to remove features with zero variance. But we also allowed for removing features with an arbitrarily small variance (in our case 0.15). We did not set out to find the optimal value for this hyper-parameter, since that would require an entire independent dataset to avoid double dipping. The layer mostly affected by this procedure was layer 7, where 7.4% of the features were removed. However, this had little to no effect on the cross-validated performance for layer 7, see below:

Here, the y-axis is our default setting used in the paper, on the x-axis we see the same results but when only zero variance features are removed. The two approaches have a resulting Pearson $r = 0.98$. Despite the small contribution of this 0.15-threshold, we would argue that it is still good practice to remove features with small variances.

Action: We have clarified the insignificant role of the hyper-parameter in the feature selection procedure on page 13-14, where we also added the link to our code:

“The prediction pipeline followed a “leave-one-image-out” procedure (i.e. train on forty-seven images and test on one left out image) – where, based on the training data, the features were thresholded to have a larger variance than 0.15, to remove near-zero-varying features, and later standardised to unit variance with a mean of zero. Our choice of a threshold of 0.15 was arbitrary and had little to no effect when compared to only removing zero variance features. It’s important to note that the test data was never part of any feature selection, as this would constitute double dipping. All pre-processing and fitting procedures were implemented using Sci-kit learn (Buitinck et al., 2013), for python code see: <https://github.com/Charestlab/abdcnn>.”

>> Figure 3A: Is the y-axis really “Proportion”, rather than the number of shuffled samples?

> Response: We thank the reviewer for pointing this out.

> Action: We have changed the label for the y-axis on figure 3A.

Figure 3A still has “Proportion.”

Response: We thank the reviewer for pointing this out.

Action: The y-label of figure 3A has now been changed to “Shuffled samples”

Reviewer #3 (Remarks to the Author):

I feel the authors have responded to my comments, and adjusted the manuscript to addresses them.

Reviewers' Comments:

Reviewer #1:

Remarks to the Author:

The authors have fully addressed my concerns.

Reviewer #2:

Remarks to the Author:

I thank again the authors for the clarification. They responded to all the point I mentioned.